# Effects of *Bacillus thuringiensis* Treatment on Expression of Detoxification Genes in Chlorantraniliprole-Resistant *Plutella xylostella*

**DOI:** 10.3390/insects15080595

**Published:** 2024-08-05

**Authors:** Maryam Zolfaghari, Fei Yin, Juan Luis Jurat-Fuentes, Yong Xiao, Zhengke Peng, Jiale Wang, Xiangbing Yang, Zhen-Yu Li

**Affiliations:** 1Guangdong Provincial Key Laboratory of High Technology for Plant Protection, Plant Protection Research Institute, Guangdong Academy of Agricultural Sciences, Guangzhou 510640, China; m.zolfaghari_89@yahoo.com (M.Z.); feier0808@163.com (F.Y.); xiaoyong@gdaas.cn (Y.X.); zkpeng0827@163.com (Z.P.); 2Key Laboratory of Green Prevention and Control on Fruits and Vegetables in South China Ministry of Agriculture and Rural Affairs, Guangdong Academy of Agricultural Sciences, Guangzhou 510640, China; 3Department of Entomology and Plant Pathology, University of Tennessee, Knoxville, TN 37996, USA; jurat@utk.edu; 4Institute of Quality Standard and Monitoring Technology for Agro-Products of Guangdong Academy of Agricultural Sciences, Guangzhou 510640, China; wangjiale@gdaas.cn; 5Subtropical Horticulture Research Station, USDA-ARS, Miami, FL 33158, USA

**Keywords:** resistance, CYP6B7, Bt-G033, RNAi, gene expression, chlorantraniliprole, diamondback moth

## Abstract

**Simple Summary:**

Detoxification genes play a crucial role in insect resistance to chemical pesticides, and exposure to biopesticides such as *Bacillus thuringiensis* (Bt) can modify their expression. Chlorantraniliprole (CAP)-resistant diamondback moth strains from China showed different expression of the detoxification genes tested (GST1, CYP6B7, and CarE-6) after treatment with CAP and Bt pesticides. The upregulation of CYP6B7 was observed after exposure to CAP, while the same gene was downregulated after larvae were exposed to Bt. Downregulation of CYP6B7 using RNAi without pretreatment with Bt resulted in increased susceptibility to CAP in resistant DBM strains, signifying a contribution of this gene to the resistant phenotype.

**Abstract:**

Detoxification genes are crucial to insect resistance against chemical pesticides, yet their expression may be altered by exposure to biopesticides such as spores and insecticidal proteins of *Bacillus thuringiensis* (Bt). Increased enzymatic levels of selected detoxification genes, including glutathione S-transferase (GST), cytochrome P450 (CYP450), and carboxylesterase (CarE), were detected in chlorantraniliprole (CAP)-resistant strains of the diamondback moth (DBM, *Plutella xylostella*) from China when compared to a reference susceptible strain. These CAP-resistant DBM strains displayed distinct expression patterns of GST 1, CYP6B7, and CarE-6 after treatment with CAP and a Bt pesticide (Bt-G033). In particular, the gene expression analysis demonstrated significant upregulation of the CYP6B7 gene in response to the CAP treatment, while the same gene was downregulated following the Bt-G033 treatment. Downregulation of CYP6B7 using RNAi resulted in increased susceptibility to CAP in resistant DBM strains, suggesting a role of this gene in the resistant phenotype. However, pretreatment with a sublethal dose of Bt-G033 inducing the downregulation of CYP6B7 did not significantly increase CAP potency against the resistant DBM strains. These results identify the DBM genes involved in the metabolic resistance to CAP and demonstrate how their expression is affected by exposure to Bt-G033.

## 1. Introduction

The diamondback moth (DBM), *Plutella xylostella* (L.), poses a significant threat to cruciferous vegetable production worldwide. Chemical pesticides have become the primary tool for DBM management [1], yet their inappropriate use and the inherent ability of DBMs to rapidly develop resistance have resulted in multiple cases of resistance. Chlorantraniliprole (CAP), an innovative anthranilic diamide insecticide that targets the ryanodine receptor (RyR) [2,3,4], has been widely used for managing lepidopteran pests, including DBM [5]. However, China, Thailand, the Philippines, and Brazil have all reported high levels of resistance to CAP in DBM populations [6,7,8]. A more effective metabolic detoxification mechanism from the increased activity or metabolic capacity of detoxification enzymes has been described in CAP-resistant insect populations [9,10,11]. In DBMs, the resistance to CAP involves increased detoxification through transport proteins [12] and metabolism by detoxifying enzymes, including members of the cytochrome P450 (CYP450), carboxylesterase (CarE), and glutathione S-transferase (GST) families [12,13,14,15,16]. 

Despite the resistance cases, CAP and other diamide insecticides remain effective tools for controlling DBM and other insect pests [17,18]. Therefore, strategies that delay resistance to CAP are highly desirable. Previous reports suggest negative cross-resistance (i.e., resistance to one pesticide resulting in increased susceptibility to an alternative pesticide) to abamectin and spinetoram in *Helicoverpa armigera* resistant to insecticidal proteins from the bacterium *Bacillus thuringiensis* (Bt) [19]. In that case, resistance to Bt proteins was linked to the lack of an ABC transporter that also reduced the detoxification of other pesticides. While the effect of Bt treatment on DBM larval susceptibility to pesticides, including CAP, has not been thoroughly studied, these Bt pesticides are commonly used in controlling DBM [20]. If DBM exposure to a Bt-based biopesticide induces reduced CAP detoxification, it could provide a valuable tool for enhancing CAP efficacy and help delay and overcome resistance. In testing this hypothesis, we identified the detoxification enzymes involved in CAP resistance in two DBM strains. We then tested the effect of the exposure to a commercial Bt pesticide used against DBM (Bt-G033) on the expression of these detoxification genes, focusing on enzymes previously shown to be upregulated in CAP-resistant DBM strains (GST 1, CYP6B7, and CarE-6) [5,13]. Pretreatment with Bt-G033 did not affect the subsequent susceptibility to CAP, suggesting a complex detoxification mechanism and a complex response to Bt-G033 in CAP-resistant DBM. The results from this study provide further insight into the mechanisms contributing to CAP resistance and advance the development of better approaches to managing DBM.

## 2. Materials and Methods

### 2.1. Insects 

A susceptible DBM (DBM-S) strain was collected in 2002 from vegetable fields in Guangdong Province, China. It has been kept isolated in the laboratory for fourteen years without exposure to insecticides. The CAP-resistant DBM strains HZ-R and GZ-R were collected in 2023 from two cabbage fields in the Guangdong Province in China (Huizhou and Guangzhu cities, respectively) with a history of excessive application of pesticides. Larvae from all strains were reared on *Brassica rapa* under laboratory conditions of 25 ± 2 °C, 65 ± 10% RH, and a 16:8 (L:D) photoperiod without pesticide exposure. The adults were fed a 10% honey solution. 

### 2.2. Insecticides 

The Bt-G033A biopesticide is based on a Bt strain expressing multiple insecticidal proteins active against lepidopteran larvae [21] and was provided by the Huazhong Agricultural University (China). CAP (50 g L^−1^ SC) was purchased from DuPont Agricultural Chemicals Ltd. (Wilmington, DE, USA).

### 2.3. Leaf-Dip Bioassays

Bioassays involved the cabbage leaf-dip method [22]. Distilled water containing 0.1% Triton X-100 (*v*/*v*) as a wetting agent was used as a diluent to prepare insecticide solutions at the required concentrations. A control (diluent) and five doses of each insecticide were tested in triplicate for each strain, with a dosage ranging between 0.003 and 0.6 ppm for Bt-G033A and 0.2 and 150 ppm for CAP. Cabbage leaf discs (6.5 cm in diameter) were made using a hole punch and then immersed into an insecticide solution for 15 s and left to air dry for 2 h at room temperature. Ten third-instar DBM larvae were used per leaf disc, and mortality based on lack of larval movement after being prodded with a fine brush was assessed after 48 h. The lethal concentration killing 50% of the individuals (LC_50_) was estimated for each pesticide and strain from experiments with <10% control mortality. 

### 2.4. Enzyme Assays

The enzymatic activity of carboxylesterases (CarEs), glutathione S-transferases (GSTs), and monooxygenase function oxidases (MFOs) was tested in vitro using homogenized third-instar DBM larvae. To create DBM homogenates, 10 larvae were homogenized in sodium phosphate buffer (0.1 M, pH 7.4) using a glass homogenizer. The resultant mixture was centrifuged at 12,000× *g* at 4 °C for 10 min, and the supernatant was used for testing enzymatic activity. The protein content was evaluated with a bicinchoninic acid (BCA) assay with bovine serum albumin as a standard, using a commercially available protein test kit (Transgen Biotech Co., Beijing, China). The CarE, GST, and MFO activities were assessed using the model substrates acetate naphetyl acetate (NA), 1-choloro-2,4-dinitrobenzene (CDNB), and phenol naphetyl acetate (p-NA), respectively [23,24,25]. Three to four enzyme assays with three technical replicates were performed for each sample. Statistical evaluations were carried out utilizing SAS version of 9.2 software [26] and the means compared with Tukey’s test statistic (*p* ≤ 0.05). 

### 2.5. Quantitative Reverse-Transcription Polymerase Chain Reaction (qRT-PCR)

The LC_10_, LC_25_, LC_50_, and LC_80_ concentrations estimated from leaf-dip bioassays for the CAP-resistant DBM strains were used in treatments of 100 larvae each. Following treatment, test insects were collected after 6, 24, and 48 h and then quickly frozen in liquid nitrogen and stored at −80 °C until used for qRT-PCR. Each of the three biological replicates performed for each concentration consisted of a pool of 5 larvae. 

Previously, a co-expression network analysis detected upregulation of three key genes (GST 1, CYP4506B7, and CarE-6) in CAP-resistant DBM strains [13]. Therefore, the expression of these three DBM genes, glutathione S-transferase 1 (GenBank: 105392138), carboxylesterase-6 (GenBank: 105382360), and PCYP6B7 (GenBank: 105380553), was investigated in response to different concentrations and time exposures to Bt-G033A and CAP. Gene-specific primers (Appendix A) were designed and synthesized by Invitrogen Trading Co. Ltd. (Shanghai, China). An EASYspin RNA isolation kit (Biomed, Beijing, China) was used to purify total RNA from each of the three pools of 5 larvae per treatment and time point, following the manufacturer’s instructions. M-MLV reverse transcriptase (Takara Bio Inc., Kusatsu, Shiga, Japan) was then used to synthesize the first-strand cDNA. Triplicate technical replicate reactions were performed in a Rotor-Gene thermal cycler (BioRad). Each 20 μL reaction mixture contained 10 μL of green qPCR superMix, 1 μL of cDNA, and 8.2 μL of ddH_2_O in addition to 0.4 μL of each primer at 0.2 M. Melting curves were analyzed to verify single amplicons, demonstrating primer specificity. Relative transcript levels were standardized by the 2^−ΔΔCT^ method [27] using the ribosomal protein RPL32 housekeeping gene (Genbank accession: AB180441) as a reference.

### 2.6. Functional Verification of CYP6B7 by RNAi

Gene-specific primers based on genomic DBM information [13] were created to amplify target CYP6B7 gene (GeneBank accession number 105380553) segments using cDNA from whole-DBM-larval-body RNA isolates as a template. Primers were designed using the SnapDragon dsRNA design tool (https://www.flyrnai.org/cgi-bin/RNAi_find_primers.pl/; accessed on 25 May 2023). A T7 polymerase promoter sequence was added to the 5′ end of the designed primers (forward primer 5′*GAATTAATACGACTCACTATAGGGAGA*CAGGGTGGTGGAAGTGAAGT-3′ and reverse primer 5′-*GAATTAATACGACTCACTATAGGGAGA* CACCTTCTTGTTTCCACCGT-3′). The following conditions were employed for the PCR: initial denaturation at 95 °C for 3 min; followed by 35 cycles of 45 s each at 95 °C, 60 °C, and 72 °C; followed by a final extension of 5 min at 72 °C. Amplicons were gel-purified using a Gel Extraction Mini Kit (Transgen) and used in an in vitro transcription system (RiboMAX T7 large-scale RNA production system, Promega, Madison, WI, USA) to synthesize dsRNA targeting the CYP6B7 gene, according to the manufacturer’s instructions. The synthesized dsRNA was treated with DNase I to remove the DNA template, precipitated with absolute ethyl alcohol, and then redissolved in RNase-free water. dsRNA targeting the green fluorescent protein (GFP) (GenBank accession number MN623123.1) was also synthesized for use as a control treatment.

Third-instar larvae received an injection of dsRNA (300 ng/larva), and CYP6B7 transcript levels were assessed by quantitative PCR 24 h later, as described elsewhere [28]. Briefly, TRIzol^®^ reagent (Invitrogen, Carlsbad, CA, USA) was used to extract total RNA from whole bodies of treated DBM larvae, which was quantified using a NanoDrop™ One microvolume UV–Vis spectrophotometer (Thermo Scientific, Wilmington, DE, USA). The quality of the purified RNA was verified using the OD260/280 ratio and observation in 1% agarose gel electrophoresis. First-strand cDNA was prepared using TransScript One-Step gDNA Removal and cDNA Synthesis SupperMix kits (Transgen). Quantitative PCR was performed using the primers detailed in Appendix A under the following conditions: denaturation at 94 °C for 30 s, followed by 40 cycles of 5 s each at 94 °C, followed by an extension of 30 s at 60 °C. The 2^−∆∆Ct^ method [27] was employed to determine the relative quantification of gene expression, with the ribosomal protein RPL32 serving as the reference gene.

At 24 h after dsRNA injection, susceptibility to CAP was tested using leaf-dipping bioassays as detailed above. Three replicates and 10 larvae in every replicate were used per treatment. An injection of dsRNA targeting the green fluorescent protein (dsGFP) was given to the control group. The PoloPlus (LeOra software, version 1, Parma, MO, USA) package was used to determine the LC_25_ from mortality data. 

### 2.7. Effects of Resistant-Strain Exposure to a Sublethal Dose of Bt-G033 

For this experiment, we used leaf bioassay test conditions from experiments outlined in Section 2.3. Twenty third-instar DBM larvae were employed per leaf disc in five replicates. Larvae were exposed to leaf discs containing the LC_10_ of Bt-G033A, and controls were exposed to leaves coated with diluent. After 6 h, the larvae from control and experimental treatments were transferred to leaf discs treated with the LC_50_ of CAP, or either water or an LC_10_ of Bt as controls. Mortality was assessed 48 h later.

## 3. Results

### 3.1. Resistance to Chlorantraniliprole (CAP) and Bt-G033A of DBM Strains

Leaf-dip bioassays were employed to assess the toxicities of CAP and Bt-G033A in three strains of DBM. The HZ-R and GZ-R strains exhibited >1000- and >500-fold resistance to CAP, respectively, compared to the DBM-S strain (Table 1). In contrast, marginal differences in susceptibility to Bt-G033A, considering natural variations in susceptibility among field populations and typical resistance levels in Bt-resistant DBM strains [29], were observed between DBM-S and either HZ-R (12-fold) or GZ-R (7-fold). These observations supported that the HZ-R and GZ-R strains were resistant to CAP but remained susceptible to Bt-G033A. 

### 3.2. Specific Activity of Detoxification Enzymes in CAP-Resistant and -Susceptible DBM Strains

The results of the enzyme activity tests using the homogenates of DBM larvae are summarized in Table 2. Both of the CAP-resistant DBM strains (GZ-R and HZ-R) had significantly higher levels of all detoxification enzymatic activities tested in comparison to the those of the susceptible (DBM-S) strain. Esterase (α-NA) activity was 2.8 and 1.9 times higher in HZ-R and GZ-R, respectively, than in the DBM-S strain. On the other hand, glutathione S-transferase (GST) activity was 3.5- and 1.6-fold higher in GZ-R and HZ-R, respectively, than in the susceptible strain. More modest differences were detected when testing for mixed-function oxidase (MFO) activity. These MFOs are cytochrome P-450-dependent intracellular enzymes, and their activity was higher in the GZ-R (1.7-fold) and HZ-R (1.2-fold) strains than in DBM-S.

### 3.3. Differences in Detoxification Enzyme Gene Expression between Strains

A complex pattern of expression was detected in members of the cytochrome P450 (CYP6B7), carboxylesterase (CarE-6), and glutathione S-transferase (GST1) detoxification gene families in response to treatment with different doses (LC_10_, LC_25_, LC_50_, and LC_80_) of CAP or Bt-G033A. Importantly, the treatments with Bt-G033A in both the GZ-R and HZ-R strains induced an overall downregulation of CYP6B7, which was overexpressed in the same CAP-resistant strains after the treatment with CAP. 

In the HZ-R strain (Figure 1), treatment with Bt-G033A induced the increased expressions of GST1 (Figure 1A) and CarE-6 (Figure 1C) in some concentrations and exposure times. Generally, the most substantial increase was detected after 6 h, followed by a reduction in expression levels after that time point. In contrast, treatment with Bt-G033A reduced the expression of CYP6B7 (Figure 1B). Overall, the expression of the three tested genes increased after the CAP treatment but followed different temporal patterns. Thus, GST1 expression increased after 6 h, then had reduced expression to a level similar to the control levels (Figure 1D). On the other hand, the expressions of both CYP6B7 (Figure 1E) and CarE-6 (Figure 1F) increased from 6 to 48 h. 

When comparing the treatments of HZ-R larvae based on the concentration of the toxicant, the sublethal (LC_10_) treatment with Bt-G033A induced increased the expressions of GST1 and CarE-6 genes, but the expression of CYP6B7 reduced, while the sublethal treatment with CAP increased the expressions of all three tested genes. Treatment with an LC_25_ dose of Bt-G033A only induced the upregulation of GST1, while treatment with an LC_50_ dose induced increased CarE-6 expression, and treatment with the highest dose (LC_80_) induced GST1 and CarE-6 expressions. Treatment with an LC_25_ of CAP increased the expression of all tested genes, an LC_50_ dose increased the expression of GST1 and CYP6B7, and an LC_80_ treatment increased dramatically the expression of CYP6B7.

Some similarities to the HZ-R response were detected in the response of the larvae from the GZ-R strain to treatment with Bt-G033A (Figure 2). For instance, treatment with Bt-G033A generally increased GST1 (Figure 2A) and reduced both CYP6B7 (Figure 2B) and CarE6 (Figure 2C) expression levels. However, CYP6B7 and CarE6 downregulation at the lowest Bt-G033A doses was more drastic in the GZ-R than in the HZ-R strain. As observed for HZ-R, increased GST1 expression was observed in GZ-R after 6 h, which decreased at subsequent time points. In contrast, treatment with CAP induced different responses in HZ-R than in GZ-R. For instance, while GST1 expression increased after CAP treatment in both strains, expression was highest in GZ-R after 48 h. Sublethal (LC_10_) treatment with Bt-G033A in GZ-R induced increased expressions of GST1 and CarE-6 genes and also the downregulation of CYP6B7 for more than 50%. In contrast to HZ-R, sublethal treatment with CAP in GZ-R increased the expression of GST1 but reduced the expressions of CYP6B7 and CarE-6 (except for the LC_10_ treatment after 6 h). Using LC_25_ or LC_50_ doses of Bt-G033A in GZ-R only induced the upregulation of GST1 at the 6 h timepoint, while an LC_80_ did not upregulate any of the tested genes. In treatments with CAP, increasing doses resulted in the increased overexpressions of GST1 and CYP6B7, but the downregulation of CarE-6. 

### 3.4. Functional Test for Participation of CYP6B7 in Resistance to CAP

Based on CYP6B7 expression being significantly increased after CAP treatment but reduced after exposure to Bt-G033A in both GZ-R and HZ-R strains, we selected this gene for further functional tests. The CYP6B7 gene was silenced using RNA interference (RNAi) and then silenced larvae were tested for susceptibility to CAP in bioassays. The injection of dsRNA reduced CYP6B7 expression by 40% compared to the injection with dsGFP in the larvae from both HZ-R and GZ-R strains (Figure 3A). In bioassays with third-instar larvae using the LC_25_ dose of CAP, the larvae from both the HZ-R and GZ-R strains with CYP6B7 knockdown displayed up to 35% higher mortality compared to the larvae injected with dsGFP (Figure 3B).

### 3.5. Sublethal Bt-G033 Pretreatment Does Not Significantly Increase Susceptibility to CAP in Resistant DBM

Based on the contribution of CYP6B7 in reducing the susceptibility to CAP and its downregulation after treatment with Bt-G033A, we tested the potential for increasing susceptibility in CAP-resistant DBM strains via the downregulation of CYP6B7 through pretreatment with Bt-G033A. The exposure of the larvae from the HZ-R and GZ-R strains to a sublethal dose (LC_10_) of Bt-G033A caused 10% and 16.7% mortality, respectively. In contrast, the exposure to an LC_50_ concentration of CAP resulted in 40% and 54% mortality in the larvae from the HZ-R and GZ-R strains, respectively (Table 3). Pre-exposure to Bt-G033A for 6 h followed by treatment with CAP for 42 h did not induce a significant difference in mortality compared to CAP treatment alone in either of the two strains tested (Table 3). 

## 4. Discussion

The effectiveness of DBM control with chemical pesticides is threatened by resistance [30,31], yet CAP and other pesticides continue to play an important role in DBM control in China. Some studies have indicated that the combined or sequential use of insecticides can increase susceptibility and reduce the resistance risk in the target pest [32,33]. The goal of the present study was to test the role of detoxification enzymes in the resistance to CAP of DBM and to test the possibility of reducing the expression of relevant enzymes through exposure to Bt-G033A.

Resistance to CAP has been previously reported in strains of DBM [13,15,34]. A point mutation in the ryanodine receptor (RyR) gene has been reported as the main allele responsible for DBM resistance to diamide insecticides [7,30,35,36]. However, the elevated activity of various detoxifying enzymes, including GST [15,37], monooxygenase [38], ABC transporters [12], and cytochrome P450 genes [5,39,40], has also been reported to be involved in resistance. The results from the enzyme assays in the present study showed an association between higher levels of GST, CarE, and MFO activities with resistance to CAP in both the HZ-R and GZ-R strains. An investigation into CAP-resistant DBM using pairwise comparisons discovered the direct participation of GST1, CYP6B7, and CarE-6 in resistance [13]. The two CAP-resistant strains examined in this study displayed differential regulation of those detoxification enzyme genes. Thus, while both GST1 and CYP6B7 showed increased expression over time after exposure to CAP, CarE-6 expression was upregulated in HZ-R but dramatically reduced in GZ-R. While speculative at this point, the upregulation of all three tested detoxification genes in HZ-R may contribute to the higher levels of resistance observed in that strain relative to GZ-R. 

The role of insect cytochrome P450 monooxygenases (CYP450), such as CYP6 genes, in the detoxification of and resistance to pesticides is well established in the literature [41]. In Lepidoptera, the overexpression of CYP6 family genes is associated with resistance to pyrethroids [42,43,44,45,46,47] and oxadiazine pesticides [48]. Previous reports indicated an association of CYP6BG1 and CYP6B6 overexpression with resistance to CAP in strains of DBM [5,39,40]. This CYP6 gene upregulation observed in CAP-resistant DBM larvae could be a result of increased expression. In this regard, *cis*-regulators have been associated with the overexpression of CYP450 genes in mosquitoes, leading to pesticide resistance [49]. Multiple *cis*-acting sequences increased the CYP6FU1 promoter activity in a strain of *Laodelphax striatellus* resistant to deltamethrin [50]. Both the FTZ-F1 transcription factor and *cis*-acting elements have been associated with the overexpression of CYP6BG1 in chlorantraniliprole-resistant DBM [39]. Further research into the regulation of CYP450 genes in DBM would be needed to identify molecular mechanisms underpinning the upregulation of CYP6B7 in CAP-resistant strains.

The contribution of CYP6B genes to resistance is supported by the results from knockdown experiments. As in the GZ-R and HZ-R strains in our study, previous reports have detected that CYP6B knockdown results in reduced resistance to CAP [5,39,40]. Similarly, the knockdown of CYP6B7 expression reduced resistance to fenvalerate in a strain of *H. armigera* [51]. Based on the observed downregulation of CYP6B7 after treatment with Bt-G033A in both CAP-resistant strains in this study, we hypothesized that pre-exposure to Bt-G033A could increase susceptibility to CAP. Previous observations already indicated increased toxicity in DBM when CAP was mixed with Bt-G033A [13]. Moreover, the two field-derived DBM strains used in this study (HZ-R and GZ-R) displayed resistance to CAP but not to Bt-G033A, supporting the lack of cross-resistance and the combined use of these products in DBM control to delay resistance evolution. However, pretreatment of CAP-resistant larvae with Bt-G033A did not increase susceptibility to CAP, thus not supporting our hypothesis. One possibility explaining these results is that multiple detoxification enzymes are needed to obtain the complete the CAP-resistance phenotype. Pretreatment with Bt-G033A pesticide downregulated CYP6B7 expression but generally had compensatory effects of increasing GST1 and CarE6 expressions. Consequently, larvae could maintain increased detoxification activity when subsequently exposed to CAP, possibly masking any effect of CYP6B7 downregulation. The findings in this study lay a foundation for further studies to identify specific genes linked to CAP resistance in DBMs and develop resistance management strategies against this pest.

## Figures and Tables

**Figure 1 insects-15-00595-f001:**
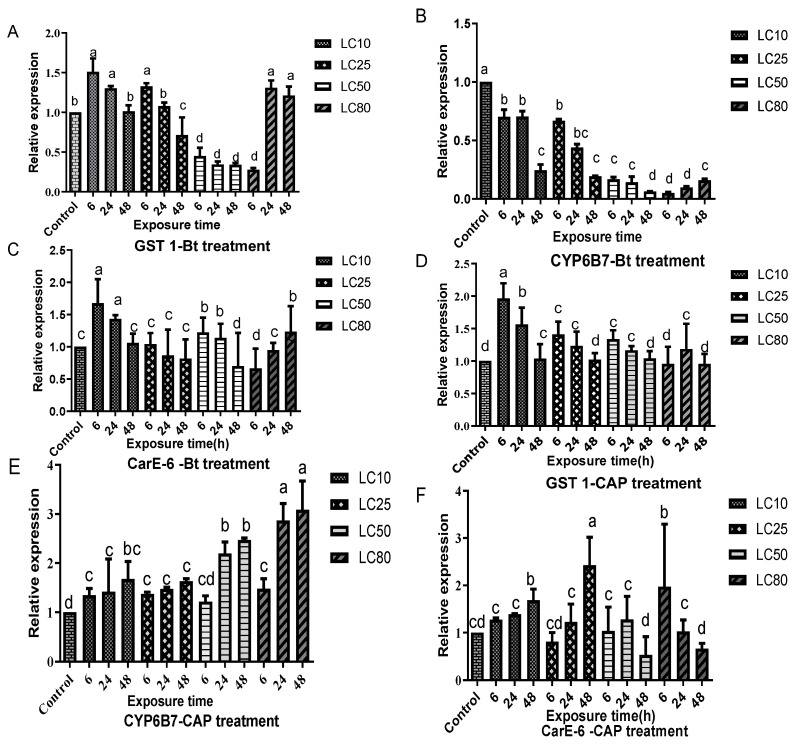
Relative expression levels of GST1 (**A**,**D**), CarE-6 (**C**,**F**), and CYP6B7 (**B**,**E**) genes in larvae from the HZ-R strain exposed to Bt-G033A (Bt-treatment) or CAP (CAP treatment) at LC_10_, LC_25_, LC_50_, and LC_80_ concentrations for 6, 24, and 48 h. Data shown are mean ± SE from three biological replicates each tested in triplicate. Different letters indicate significant differences (Tukey’s test, *p* < 0.05).

**Figure 2 insects-15-00595-f002:**
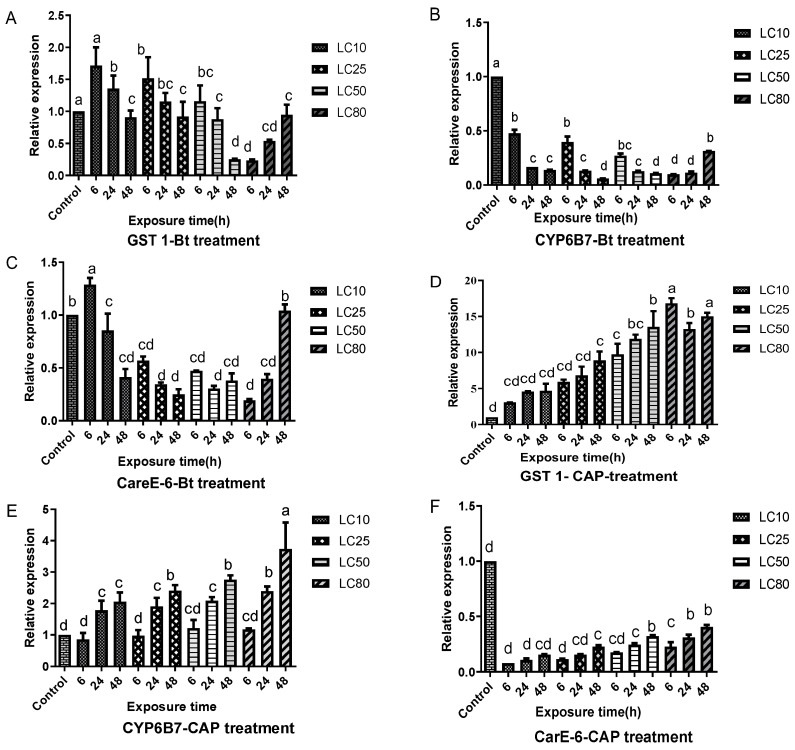
Relative expression levels of GST1 (**A**,**D**), CarE-6 (**C**,**F**), and CYP6B7 (**B**,**E**) genes in larvae from the GZ-R strain after exposure to Bt-G033A (Bt treatment) and CAP (CAP treatment) at LC_10_, LC_25_, LC_50_, and LC_80_ concentrations for 6, 24, and 48 h. Data shown are mean ± SE from three biological replicates tested in triplicate. Different letters indicate significant differences (Tukey’s test, *p* < 0.05).

**Figure 3 insects-15-00595-f003:**
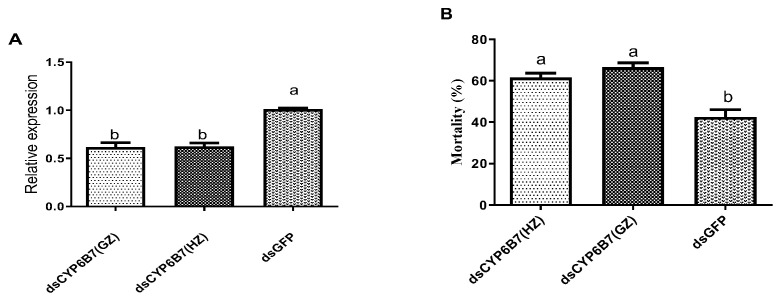
Relative expression levels of CYP6B7 (**A**) and percentage mortality (**B**) in larvae from the GZ and HZ strains after CYP6B7 knockdown by RNAi. (**A**) Data shown are mean ± SE from three biological replicates (individual whole larvae) tested in triplicate. (**B**) Bioassays were performed with larvae injected with dsRNA targeting CYP6B7 or GFP (control) and then treated with an LC_25_ of CAP one day after injection. Mortality was determined after three days. Data shown are mean ± SE from 5 replicates of 100 larvae for each strain. The significance of differences between control (dsGFP) and treatments was analyzed by a *t*-test (*p* < 0.05) and is denoted by different letters on each column.

**Table 1 insects-15-00595-t001:** Toxicity of chlorantraniliprole (CAP) and Bt-G033A against three strains of DBM.

Insecticide	Population	N ^a^	LC_50_ (μg/mL)	95% CI ^b^	χ2 (df)	RR ^c^
CAP	DBM-S	150	0.05	0.044–0.056	0.9(3)	1
	GZ-R	150	27.4	21.3–34.8	0.8(3)	548
	HZ-R	150	52.84	45.67–62.37	0.7(3)	1057
Bt-G033A	DBM-S	150	0.012	0.01–0.0142	0.7(3)	1
	GZ-R	150	0.08	0.07–0.088	0.8(3)	7
	HZ-R	150	0.15	0.1–0.25	0.7(3)	12

^a^ N = number of insects tested per strain and insecticide. ^b^ CI = confidence interval. ^c^ RR: resistance ratio = LC_50_ of the resistant strain/LC_50_ of the susceptible strain.

**Table 2 insects-15-00595-t002:** Esterase, glutathione S-transferase (GST), and mixed-function oxidase (MFO) activities in homogenates of larvae from three different DBM strains. Shown are mean specific activity values from 5 replicates using homogenates of 10 larvae each. Activities were measured using α-NA as a substrate for esterase, 1-chloro-2,4-dinitrobenzene (CDNB) for GST, and ρ-NA for MFO. Significant differences (*p* < 0.05, Tukey’s test) between means in a column are denoted by a different letter.

Strain	Esterase Mmol/min/mg Protein ± SE	^a^ AR	GST Mmol/min/mg Protein ± SE	AR	MFO nmol/min/mg Protein ± SE	AR
DBM-S	0.014 ± 0.005 a	1	0.12 ± 0.01 a	1	3.9 ± 0.02 a	1
GZ-R	0.027 ± 0.001 b	1.9	0.43 ± 0.03 b	3.5	5.6 ± 0.04 b	1.4
HZ-R	0.040 ± 0.003 c	2.8	0.20 ± 0.03 c	1.6	4.8 ± 0.01 c	1.2

^a^ AR: activity ratio = activity in the resistant strain/activity in the susceptible strain.

**Table 3 insects-15-00595-t003:** Mortality of third-instar DBM larvae from two CAP-resistant strains after exposure to water or an LC_10_ of Bt-G033A (Bt) for 6 h followed by exposure for 42 h to an LC_50_ of CAP. The data are expressed as the mean percentage mortality ± SE from 5 bioassay replicates with 20 DBM larvae each. Different letters within a column indicate significant differences as determined by the Duncan test (*p* < 0.05).

Strain	Treatment	Larval Mortality (%)
GZ-R	Bt + CAP	45.0 ± 1.6 ab
	Water + CAP	54.0 ± 1.9 b
	Bt	16.7 ± 6.7 c
	Water	6.7 ± 3.3 de
HZ-R	Bt + CAP	45.0 ± 1.6 ab
	Water + CAP	40.0 ± 2.2 b
	Bt	10.0 ± 5.8 cd
	Water	0.0 ± 0.0 e

## Data Availability

The raw data supporting the conclusions of this article will be made available by the authors without undue reservation.

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
