# Peer review of "Effects of Bacillus thuringiensis Treatment on Expression of Detoxification Genes in Chlorantraniliprole-Resistant Plutella xylostella"

_insects, 2024, doi:10.3390/insects15080595_

Round 1

Reviewer 1 Report (New Reviewer)

Comments and Suggestions for Authors

The authors used RNAi to knock down the expression of CYP4506B7. Could the authors give the details about how they design RNAi? The regular RNAi experiment always design several dsRNAs to knock down specific gene. Please explain.

Author Response

Reviewer: 1

Response: We truly appreciate the reviewer’s encouraging remarks and constructive comments on the manuscript. Following is our response to the reviewer’s comments/concerns.

The authors used RNAi to knock down the expression of CYP4506B7. Could the authors give the details about how they design RNAi? The regular RNAi experiment always design several dsRNAs to knock down specific gene. Please explain

We designed several specific primers for the target gene CYP6B7 using the SnapDragon dsRNA design tool (https://www.flyrnai.org/cgi-bin/RNAi_find_primers.pl) with DBM genomic information as a template. The efficiency of each primer was assessed, and the primer with the highest efficiency for the CYP6B7 gene was selected. This information is now detailed in the Materials and Methods section. While we designed multiple primers, the set with the most specific amplification was used to prepare dsRNA, the only dsRNA we tested. We acknowledge and agree with the reviewer that preparing and testing multiple dsRNAs targeting the same gene helps increase the probability of knockdown success. However, in our case, we observed significant silencing with the synthesized dsRNA that allowed testing of our hypothesis, so no additional dsRNAs were tested.

Reviewer 2 Report (New Reviewer)

Comments and Suggestions for Authors

The authors describe the characterization of some genes that have been described to participate in possible resistance mechanisms to the pesticide chlorantraniliprole (CAP) in Plutella xylostella larvae. Based on previous work, they focused on the expression of CYP6B7, which increases its expression in CAP-resistant larvae, but its expression is reduced when the larvae are exposed to sublethal concentrations of Bacillus thuringiensis. Although the article is well structured, there are some details that should be clarified by the authors: In figures 1 and 2: The fillings of the bars do not seem to coincide with the LCs values, could you verify this? They could indicate in the text how much the expression of CYP6B7 is reduced after exposing the larvae for 6 h to the LC10 of BT. If I understood correctly, both in the silencing experiment and in exposure to Bt (LC10, 6h) the expression of CYP6B7 is reduced by approximately 50%? Did you try any dose(s) other than LC25 of CAP in your silenced larvae? In silenced larvae, does the LC50 value of CAP decrease?

Author Response

Reviewer: 2

Response: We truly appreciate the reviewer’s encouraging remarks and constructive comments on the manuscript. Following is our point-by-point response to the reviewer’s comments/concerns.

In figures 1 and 2: The fillings of the bars do not seem to coincide with the LCs values, could you verify this? 

Response: Thank you for bringing this to our attention. We have reviewed Figures 1 and 2 and confirmed that the fillings of the bars do not accurately reflect the LCs values. We apologize for any confusion this may have caused. The figures have been revised to ensure alignment between the bar fillings and the LCs values.

They could indicate in the text how much the expression of CYP6B7 is reduced after exposing the larvae for 6 h to the LC10 of Bt

Response: Thank you for your suggestion. The expression of CYP6B7 was reduced by more than 50% after exposing the DBM larvae for 6 h to the LC10 of Bt. We have incorporated this information into the manuscript (Results, section3.3, paragraph 3)

If I understood correctly, both in the silencing experiment and in exposure to Bt (LC10, 6h) the expression of CYP6B7 is reduced by approximately 50%? Did you try any dose(s) other than LC25 of CAP in your silenced larvae? In silenced larvae, does the LC50 value of CAP decrease?

Response: We appreciate the reviewer's thorough examination of our experimental data and the opportunity to clarify the details.

Treatments with Bt-G033A in the HZ-R strain with ~1,000-fold resistance to CAP reduced 50-80% the expression of CYP6B7 at different time points and in all Bt concentrations tested (LC10, LC25, LC50, and LC80).  In the GZ strain with ~500-fold resistance to CAP treatment with Bt-G033A resulted in between 30-80% reduced CYP6B7 expression in most tested concentrations and time points. Knockdown for both strains treated with the LC25 of Bt-G033A was about 40%. (data from 5 replicates of 100 DBM larvae each per strain). We also performed test treatments with the LC50 concentration and observed CAP down-regulation, but we, unfortunately, lost those resistant populations and were left with only the replicated LC25 data presented in the manuscript. We did not perform dose bioassays to estimate the LC50 for CAP against CYP6B7 silenced larvae, but data in Fig. 3B shows that resistance was significantly reduced in these larvae compared to the control.

This manuscript is a resubmission of an earlier submission. The following is a list of the peer review reports and author responses from that submission.

Round 1

Reviewer 1 Report

Comments and Suggestions for Authors

The manuscript presented by Maryam Zolfaghari and collaborators state that Plutella xylostella decreases the expression of cytochrome CYP6B7 upon treatment with Bacillus thuringiensis strain G033. This downregilation contributes to a detoxification of the chemical insecticide Chlorantraniliprole, leading to higher susceptibility in P. xylostella. In my opinion, data on downregulation of CYP6B7 by Bt G033 and upregulation by CAP seems clear. However, the bioassay results are week and do not support the hypothesis. RNAi assay decreases the CYP6B7 expression in 30%, leading to 35% of higher mortality. These results should be reconsidered because down-regulation of CYP6B7 is low and an effort to obtain higher rate of RNA silencing should be obtained. Furthermore, the RNA silencing should be tested with individual larvae in order to check the average of larvae silenced. Furthermore, if downregulation of CYP6B7 contributes to CAP susceptibility, then one expects that pre-treatment with BtG033 should increase the insect susceptibility, but table 3 do not provide this phenotype.

Reviewer 2 Report

Comments and Suggestions for Authors

This manuscript by Zolfaghari et al. explores the potential for Bt to be used as a potentiator for the diamide insecticide, chlorantraniliprole (CAP). Initially, expression of a trio of metabolic resistance genes and their protein products was studied in two CAP resistant lines of DBM, revealing significant alteration of expression by exposure to both Bt and CAP. The most significant effect was found to be reduction of the CYP by exposure to Bt and further work focussed on this gene. RNAi silencing of the CYP increased CAP toxicity, but pre-treatment with Bt did not. The studies reveal some interesting results, but both their presentation and interpretation are flawed in numerous ways.

The biggest problem stems from the interpretation of the Bt pre-exposure data. Although the data in Table 3 show that pre-exposure to Bt did not induce a significant increase in mortality after subsequent treatment with CAP for either of the two strains tested, the title of this section (3.5) inexplicably says that it did. Even more surprisingly, the same incorrect claim is made in the Abstract on lines 31-32. This significantly erodes the reader’s faith in the reliability of the interpretations of the results presented.

Other cases of inaccurate claims are made elsewhere in the manuscript, including:

Lines 14-15 - chlorantraniliprole (CAP)-resistant diamondback moth strains from China WERE NOT shown to have higher expression of detoxification genes (GST1, CYP6B7, CarE-6) compared to susceptible strains in this manuscript.

Lines 22-25 – the same incorrect claim is made again in the Abstract. General enzyme assays were used which do not specifically measure enzyme levels resulting from expression of the three genes listed.

Lines 204-205 - the patterns of alteration in expression to the tested genes are complex. Despite the problems with bar labeling, it is still generally evident what the bars represent, yet the interpretations of the results by the authors are at times questionable. For example, on lines 204-205, the authors suggest Bt treatment increased expression of GST1 (Fig 1A), yet there are an equal number of conditions decreased as there are increased. The temptation to make generalizations might better be minimized for these figures.

In some instances, more information would be beneficial, including:

Line 79 - how long were the R strains maintained in the absence of insecticide pressure?

Line 108 - what were the replicate numbers used?

Lines 128-130 – what is the evidence that this reference gene is valid?

Lines 141-143 - The method for dsRNA production is not well explained, please clarify.

Lines 145-147 - More details are needed about what "both” the CYP6B7 transcripts" are. If this gene has alternative transcripts, please give some details about how it is being measured.

Lines 252-254 - this is very weak silencing. Were multiple targets in this gene tried? Better silencing might have produced a stronger result.

Figure 3B - mortality of the control group when treated with an LC25 is over 40%. Is this because the insects have been injured with an injection the day before CAP exposure?

Lines 255 and 269 - why was the CAP exposure done with an LC50 dose for Bt pre-treatment, but with an LC25 dose for RNAi gene silencing?

There are also a number of issues with presentation for the manuscript, some more significant than others. There are typos throughout the text as well as grammatical errors. Structural issues include:

Line 107 – there is a problem here with the reference software.

Lines 137-138 – are Transgene and Transgen,China the same supplier?

Line 144 - please provide the supplier for the Ampliscribe kit.

Line 146 - why is this not a reference in the list?  

Lines 163-168 – The entire section 2.7 needs to be rewritten.

Figures 1 and 2 - all the legend patterns within the figures are out of phase with the bar graphs. Patterns are also inconsistent within the bars of the graphs. I would suggest the authors use color instead of patterns for different conditions. The authors should also check the letters indicating significant differences as these appear to have some inaccuracies (for example, panels 1D, 2A, 2F). Labels of axes are also inconsistent.

Line 256 – there is no Fig 4.

Comments on the Quality of English Language

There are typos throughout the text as well as grammatical errors. Most of these could be easily caught by a proofreading from English speaking authors.